# Numerical Simulation Study on Predicting the Critical Icing Conditions of Aircraft Pitot Tubes

**DOI:** 10.3390/s24227410

**Published:** 2024-11-20

**Authors:** Qixi Chen, Lifen Zhang, Chengxin Zhou, Zhengang Liu, Yaguo Lyu

**Affiliations:** School of Power and Energy, Northwestern Polytechnical University, Xi’an 710072, China; chen_qixi404@mail.nwpu.edu.cn (Q.C.); zcx21zrx99@mail.nwpu.edu.cn (C.Z.); zgliu@nwpu.edu.cn (Z.L.); yglu@nwpu.edu.cn (Y.L.)

**Keywords:** pitot tube icing, critical icing conditions, numerical simulation

## Abstract

Aircraft pitot tubes are sophisticated instruments designed to detect airflow pressure and relay this information to onboard computers and flight instruments, enabling the calculation of airspeed through the measurement of total-static pressure differences. The formation of ice on aircraft pitot tubes can compromise the acquisition of airspeed data, misguide pilots, and potentially cause catastrophic flight control failures. This article introduces a predictive methodology for identifying critical conditions that lead to icing on aircraft pitot tubes. Utilizing numerical simulation techniques, the methodology calculates the critical conditions for pitot tube icing across cruise flight regimes and atmospheric conditions, resulting in the generation of a critical condition envelope surface. By comparing these critical conditions against actual sensor data, a predictive danger zone can be established, offering an advanced warning system to ensure flight safety.

## 1. Introduction

When an aircraft passes through clouds containing supercooled water droplets or encounters freezing rain, icing may occur on the windward side. This article focuses on the discussion of aircraft pitot tube icing as is shown in Figure 1. An aircraft pitot tube is a sensor that obtains airspeed data by sensing and calculating the difference between total pressure and static pressure. Ice accumulation on the surface of the aircraft pitot tube may cause airspeed data distortion and affect flight safety [1]. Civil aviation aircraft rely on data provided by the aircraft pitot tube for operation, and airspeed data distortion can affect flight judgment, even leading to autopilot detachment and flight control system failure, resulting in accidents.

According to reports from the European Aviation Safety Agency and the National Aeronautics and Space Administration of the United States, flight safety accidents caused by icing of various aircraft components have occurred frequently in the past 20 years, and the data is shown in Figure 2 [3]. Among them, accidents caused by the icing of pitot tubes are particularly serious. In 2009, the autodrive system of Air France Flight F447 failed because of the icing of the pitot tube, resulting in 288 deaths [2]. In 2018, the Russian An-148 aircraft also crashed for the same reason, resulting in 71 deaths [4]. In addition, in 1997, China Southern Airlines Flight 2553 crashed into the ground because of the icing of the aircraft pitot tubes [2]. Although modern aircraft and weather radars are capable of offering even more precise forecasts of flight conditions, the rapid development of commercial aviation still makes icing accidents severe. Since 1973, at least 13 icing accidents related to aircraft pitot tubes have resulted in 383 deaths [2].

It is evident that current technology struggles to address the issue of icing during aircraft flight completely. Given the high risk of severe safety incidents resulting from icing on aircraft pitot tubes, it is imperative to investigate the icing safety margins and critical conditions for the operation of aircraft pitot tubes to ensure flight safety.

In recent years, researchers from various countries have gradually initiated a variety of forms and directions of research on the icing and anti-icing of aircraft pitot tubes.

De Souza et al. validated a convective CHT model for transient heat conduction in aircraft pitot tubes and integrated it into the heat transfer control equation for icing numerical calculations, thereby enhancing the accuracy of icing prediction in numerical calculation processes [5], which was used for the design and optimization of anti-icing systems [6]; Jackel et al. developed and validated a numerical model to investigate the surface temperature distribution of aircraft pitot tubes with varying internal compositions and material properties in the event of an anti-icing system failure [7]. They integrated this outcome with the CHT model to devise a groundbreaking redundant heating system specifically for aircraft pitot tubes, one that incorporates Phase Change Materials (PCMs) [8]. In addition, Rodrigues et al. proposed a method based on acoustics to detect whether the aircraft pitot tubes are blocked [9].

Furthermore, numerous researchers have undertaken various studies to enhance the precision of water droplet impact characteristics in numerical icing calculations. These studies involve initially calculating the water droplet impact velocity and angle, then determining the local collection coefficient, and ultimately employing the MOF multiphase flow method to simulate the impact process [7] or Jung and Myong et al., among others, have converted non-strict hyperbolic equations into strict hyperbolic ones by incorporating artificial pressure terms and by introducing approximate Riemann solvers to enhance the accuracy of water droplet impact calculations [8,9,10,11].

To optimize thermodynamic models for ice formation and water film flow, research has primarily focused on developing ice formation models that more accurately reflect actual ice formation and engineering scenarios. This effort aims to address the deficiency of the Messinger ice formation model in handling the overflow of surface water. Yi Xian et al. proposed a three-dimensional ice formation calculation model that takes into account the overflow water effect of liquid water content and established a distribution scheme for overflow water flow within surface units [10]. Chang Shinan et al. used the custom function of Fluent software to extend the two-dimensional Messinger model to three dimensions and completed a numerical simulation of wing icing [11]. Kong and Liu established a super-cooled ice formation model based on the growth characteristics of ice crystals in super-cooled water and proposed a multistage ice formation model [12].

With the significant development of computer and CFD technology, an increasing number of research institutions and enterprises have developed a series of specialized icing calculation software, such as LEWICE from the United States [13,14,15,16,17], FENSAP-ICE from Canada [18,19,20,21], ONERA from France, TRAJICE2 from the UK, CIRAAMIL from Italy, etc. LEWICE and FENSAP-ICE are currently the two most widely used and representative types of numerical simulation software for icing.

If the critical icing conditions that could lead to aircraft pitot tube failure during the climb, cruise, and landing phases are identified and fed into the flight computer, environmental data can be derived from meteorological radar and temperature sensors. Additionally, flight status data can be gathered from aircraft pitot tubes and angle of attack sensors, among others. By performing calculations through these above data, icing events can be anticipated, and ECAM warnings can be generated by the flight warning computer to alert pilots. This proactive measure aims to prevent airspeed data failure, minimize the risk of operational errors, and thereby ensure flight safety [22].

Based on the aforementioned considerations, this article employs numerical simulation to model the icing process of the A320-200 aircraft pitot tube during cruise conditions. It identifies the critical icing conditions that could lead to pitot tube failure and establishes a safe flight envelope accordingly. The research concepts and methodologies presented herein can, to a certain extent, decrease the likelihood of operational errors by aircraft autopilots and pilots during icing conditions, thereby enhancing the safety of civil and commercial aviation operations.

## 2. Model of Numerical Simulation Calculation

### Computation Model

The calculation of airfield and water-droplet impact characteristics primarily employs unstructured grids, utilizing the classical Galerkin Finite Element Method to solve the Reynolds-Averaged Navier–Stokes (RANS) equations, the Navier–Stokes (N-S) equations [1], the turbulence model partial differential control equations, and the Euler-described air and supercooled water droplet two-phase flow control equations.
Mass-conservation equation:
Regarding the air:
(1)$$\frac{\partial \rho }{\partial t}+\frac{\partial }{\partial x_{i}}\left(\rho u_{i}\right)=0$$
Regarding supercooled water droplets:
(2)$$\frac{\partial \alpha }{\partial t}+\vec{\nabla }\times \left(\alpha {\vec{u}}_{d}\right)=0$$
2.Momentum conservation equation:
Regarding the air:
(3)$$\frac{\partial }{\partial t}\left(\rho u_{i}\right)+\frac{\partial }{\partial x_{j}}\left(\rho u_{j}u_{i}\right)=-\frac{\partial p}{\partial x_{i}}+\frac{\partial \sigma_{ij}}{\partial x_{j}}+\frac{\partial }{\partial x_{j}}\left(\rho u_{i}\prime u_{j}\prime \right)$$
Regarding the supercooled water droplets:
(4)∂αV→d∂t+∇→×αu→du→d=CDRed24KIαu→a−u→d+α1−ρaρd1Fr2g→
3.Energy conservation equation:
Regarding the air:
(5)$$\frac{\partial }{\partial t}∭uWdV+∯sFdS=0$$
Regarding the supercooled water droplets:
(6)$$\frac{\partial \rho_{a}E_{a}}{\partial t}+\vec{\nabla }\times \left(\rho_{a}{\vec{u}}_{a}H_{a}\right)=\vec{\nabla }\times \left[\kappa_{a}\left(\vec{\nabla }T_{a}\right)+\nu_{i}\tau^{ij}\right]+\rho_{a}\vec{g}\times {\vec{u}}_{a}$$
4.Differential control equation of turbulence model:
S-A model [23]:Transport equation:
(7)$$\frac{\partial }{\partial t}\left(\rho \tilde{\nu }\right)+\frac{\partial }{\partial x_{i}}\left(\rho \tilde{\nu }u_{i}\right)=G_{v}+\frac{1}{\sigma_{\tilde{v}}}\left[\frac{\partial }{\partial x_{j}}\left\{\left(\mu +\rho \tilde{\nu }\right)\frac{\partial \tilde{\nu }}{\partial x_{j}}\right\}+C_{b2}\rho {\left(\frac{\partial \tilde{\nu }}{\partial x_{j}}\right)}^{2}\right]-Y_{v}+S_{\tilde{v}}$$
Energy equation:
(8)$$\frac{\partial }{\partial t}\left(\rho E\right)+\frac{\partial }{\partial x_{i}}\left[u_{i}\left(\rho E+p\right)\right]=\frac{\partial }{\partial x_{i}}\left[\left(k+\frac{c_{p}u_{t}}{{\mathrm{Pr}}_{t}}\right)\frac{\partial T}{\partial x_{j}}+u_{i}{\left(\tau_{ij}\right)}_{eff}\right]+S_{h}$$
in which:(9)$${\left(\tau_{ij}\right)}_{eff}=\mu_{eff}\left(\frac{\partial u_{j}}{\partial x_{i}}+\frac{\partial u_{i}}{\partial x_{j}}\right)-\frac{2}{3}\mu_{eff}\frac{\partial u_{k}}{\partial x_{k}}\delta_{ij}$$
k-ω model [24]: It introduces the turbulent kinetic energy *k* and the specific dissipation rate *ω*, which are based on the RANS equations, to obtain its control equations:(10)$$\frac{\partial k}{\partial t}+\nabla \cdot \left(k\vec{u}\right)=\nabla \left(\left(\nu +\mu \sigma_{k}\right)\nabla_{k}\right)+{\vec{\tau }}_{j}:\nabla \vec{u}-\beta^{*}\omega k$$
(11)$$\frac{\partial \omega }{\partial t}+\nabla \left(\omega \vec{u}\right)=\nabla \cdot \left(\left(\nu +\mu \sigma_{\omega }\right)\nabla_{\omega }\right)+\alpha \frac{\omega }{k}\tau_{r}:\nabla \vec{u}-\beta \omega^{2}$$
where the “:” operator in the equation is used to define the matrix inner product, it contains five closure coefficients:

$\sigma_{k}\in \left[0,1\right]$; $\sigma_{\omega }\in \left[0,1\right]$; $\frac{\beta^{*}}{\beta }\in \left[0.9,2.5\right]$; $\beta^{*}\in \left[0.0292,0.2034\right]$; $\beta \in \left[0.01168,0.226\right]$; $\alpha \in \left[0,1.11\right]$.
5.Control equation for two-phase flow of air-supercooled water droplets:(12)$$\nabla \cdot \left(\rho_{d}\alpha {\vec{u}}_{d}\right)=0$$(13)$$\nabla \left(\rho_{d}\alpha {\vec{u}}_{d}{\vec{u}}_{d}\right)=\rho_{d}\alpha K\left({\vec{u}}_{a}-{\vec{u}}_{d}\right)+\rho_{d}\alpha G_{d}$$

The control equations for icing calculation are as follows:

Partial differential equation of water film velocity:(14)$$\bar{u}\left(x\right)=\frac{1}{h_{f}}\int_{0}^{h_{f}}u\left(x,y\right)dy=\frac{h_{f}}{2\mu_{w}}\tau_{wall}\left(x\right)$$
Conservation of mass equation:
(15)$$\rho_{w}\left[\frac{\partial h_{f}}{\partial t}+div\left(\bar{u}h_{f}\right)\right]=U_{\infty }LWC\beta -{\dot{m}}_{evap}-{\dot{m}}_{ice}$$
Energy conservation equation:
(16)$$\begin{array}{l} \rho_{w}\left[\frac{\partial h_{f}C_{w}\tilde{T}}{\partial t}+div\left(\bar{u}h_{f}C_{w}\tilde{T}\right)\right]=\left[C_{w}{\tilde{T}}_{d,\infty }+\frac{{\left\|u_{d}\right\|}^{2}}{2}\right]\times U_{\infty }LWC\beta \\ -0.5\left(L_{evap}+L_{subI}\right){\dot{m}}_{evap}+\left(L_{fusion}-C_{ice}\tilde{T}\right){\dot{m}}_{ice}+\sigma \left(T_{\infty }^{4}-T^{4}\right)+{\dot{Q}}_{h}+{\dot{Q}}_{cond} \end{array}$$

## 3. Numerical Calculation of Icing

### 3.1. Calculation Process

The numerical simulation of critical icing conditions, as detailed in this article, primarily encompasses the following steps, as illustrated in Figure 3: three-dimensional modeling of the object under study; dividing the constructed 3D model into unstructured grids and defining the fluid domain; conducting airfield calculations within the defined fluid domain; calculating the impact characteristics of water droplets based on the outcomes of the airfield calculations; performing icing prediction calculations based on the characteristics of water droplet impact and outputting the grid post-icing; recalculating the airfield of the grid post-icing to ascertain the effect of icing on the surrounding airflow field of the object under study; iterating the calculations by varying the icing conditions.

### 3.2. Research Object

This article undertakes numerical simulation to predict the critical conditions leading to failure and icing of aircraft pitot tubes, with a specific focus on the 0851HL pitot tube installed on A320 aircraft as the research subject. By constructing a detailed three-dimensional model, the subsequent 3D representation which is shown in Figure 4 is derived:

### 3.3. Grid Division and Fluid Domain Division

The computational domain (which is shown in Figure 5) has been subdivided into tetrahedral grids, with enhanced mesh density around the inlet horn tube and overflow hole, as depicted in Figure 6. The overall cell count amounts to 277,502.

### 3.4. Boundary

The boundary conditions of the computational domain are illustrated in Figure 7, with both the inlet and outlet configured as pressure-far-field boundary conditions. These conditions are defined by a temperature of −41.15 °C, a pressure of 33,161.2 Pa, and an incoming Mach number of 0.752, and the boundary conditions and their corresponding values are displayed in Table 1. The remaining walls are considered insulated walls with a roughness of 0.05 mm.

### 3.5. Example Verification

The icing results of the NACA0012 airfoil in reference [25] were used to verify the correctness of the calculation method proposed in this paper. The validation example parameters and conditions are:

Wing chord length: 0.5334 m.

The calculation conditions are shown in the Table 2:

The comparison of the calculated results with the ice shapes depicted in the literature is illustrated in Figure 8.

As illustrated in Figure 8, the ice growth trend derived from our calculations aligns well with the experimental outcomes, and the calculated icing threshold corresponds precisely with these empirical data. The maximum ice thickness is also quite proximate to the experimental findings. Consequently, it is deemed that the numerical calculation method presented in this paper is both feasible and effective.

## 4. Prediction of Critical Conditions for Aircraft Pitot Tube Failure and Icing

The numerical calculation method for icing, as outlined in the preceding text, has been employed to assess the icing conditions of aircraft pitot tubes computationally. Following the analysis of the calculated results, the critical condition envelope surface for the failure icing of the aircraft pitot tube has been derived.

### 4.1. Calculation Conditions

This article investigates the icing conditions affecting the pitot tube on A320 aircraft, focusing primarily on the conditions encountered during cruise flight (which had been shown in Table 3) at an altitude of 8500 m and a speed of 0.752 Mach. When setting the icing conditions, please refer to the temperature and pressure of the region. Calculate the duration of different icing conditions, the LWC (ranging from 0.1 to 1 g/m^3^), and the MVD (ranging from 10 to 50 μm) to study the icing conditions of the aircraft pitot tube. Additionally, the icing times calculated under different LWC and MVD are given in Table 4, Table 5 and Table 6. Calculate the error between the total pressure measured by the pitot tube after icing and the actual total pressure and plot the curve of total pressure error versus icing duration under different icing conditions. Set the maximum allowable range and determine the critical icing conditions that lead to the failure of the aircraft pitot tube.

In summary, this article computes icing by utilizing the variable values associated with the characteristic states within the range of the examined states:

### 4.2. Calculation Results

#### 4.2.1. Calculation Results of Airfield and Water Droplet Impact Characteristics

Initially, we compute the airflow field with the outcomes depicted in Figure 9. It is evident that:The turbulence intensity at the inlet horn and the leading edge of the aircraft pitot tube’s support plate is relatively low. At these points, incoming water droplets do not acquire additional directional velocity from the turbulence and instead collide directly with the low-temperature surface, leading to ice formation. This process reduces the likelihood of water droplets overflowing.The temperature at the inlet of the aircraft pitot tube and the leading edge of the support plate is below −10 °C, indicating that icing conditions are present at this location.

On the basis of the airfield calculation results, numerical calculations of water droplet impact characteristics were carried out, and the results are shown in Figure 10 and Figure 11. It can be seen that:The pitot tube exhibits the highest rate of water droplet collection at its inlet horn aperture and the front of the support plate. Field data (refer to Figure 9a) indicates that temperatures at these points are relatively low, rendering them susceptible to icing.When the MVD remains constant, and the LWC rises, there is a significant increase in the rate of water droplet collection. Conversely, if the LWC stays the same and the MVD increases, the rise in the rate of water droplet collection is not significant.

#### 4.2.2. Calculation Results Ice Formation and Flow Field After Icing

Upon conducting numerical simulations of icing based on the calculated parameters of the airflow and water droplet distribution, as depicted in Figure 11, it was observed that as the duration increased, the accumulation of ice at the inlet of the aircraft pitot tube progressively escalated. Consequently, the pitot tube became increasingly obstructed. This obstruction prevented the detection of total pressure data, ultimately causing a failure (Here, the calculation results for ice accumulation are presented exclusively under the condition of LWC = 0.1 g/m^3^ and MVD = 50 microns, which has been shown in Figure 12).

Table 7 illustrates the critical phenomenon of ice blockage in pitot tubes. It is evident that as the LWC and the MVD increase, the duration of ice blockage progressively decreases.

Import the frozen surface into the N-S solver and perform numerical calculations of the airfield again to obtain total and static pressure data sensed by the aircraft pitot tube after icing. Processing these data can obtain the error between airspeed data output by the aircraft pitot tube and actual airspeed data as a function of icing time, MVD, and LWC in the air.

Define the aircraft pitot tube error as follows:(17)$$error=\frac{\left|u_{Flow}-u_{sensor}\right|}{u_{Flow}}\times 100\%$$
where

*u_Flow_*—Actual flow speed;*u_sensor_*—The airspeed sensed by the pitot tube.

When the MVD of water droplets remains constant, the relationship between the error in aircraft pitot tube output airspeed data and the LWC and icing time is depicted by the curves in Figure 13. These curves are represented by clusters of lines in red (MVD = 20 microns), blue (MVD = 35 microns), and green (MVD = 50 microns).

When the LWC of water droplets remains constant, the error in airspeed data output by the aircraft pitot tube fluctuates in accordance with the MVD of the droplets and the duration of icing. This is illustrated by the box-solid line cluster (LWC = 0.1 g/m^3^), circle-dashed line cluster (LWC = 0.5 g/m^3^), and triangle-dotted line cluster (LWC = 1 g/m^3^) in Figure 13.

### 4.3. Data Analysis

The chart analysis shows that during cruising flight when other conditions remain constant:(1)When LWC remains constant, as MVD increases, given the same icing duration, the airspeed error induced by the pitot tube icing exhibits an upward trend. However, the impact of water droplets and the icing process is inherently complex, involving a multi-parameter coupling in its numerical computation. Consequently, during the actual calculation process, it becomes evident that under certain specific icing conditions (such as those represented by the blue circle-solid line and blue circle-dashed line in Figure 13, where MVD is at an intermediate value of 35 microns), the time required for the pitot tube to become completely obstructed by ice is longer than under other conditions. This results in a greater accumulation of ice within the pitot tube channel when blocked, leading to a more significant error at the moment of complete blockage compared to other conditions. Simultaneously, with MVD held constant, the rate at which the airspeed error increases with rising LWC becomes notably steep.(2)When the MVD remains constant, an increase in LWC leads to a trend of escalating errors in airspeed measurements due to pitot tube icing over the same icing duration. Nevertheless, water droplet impingement and icing constitute a complex physical process, and its numerical simulation is a multi-parameter coupled process. Consequently, in practical calculations, it becomes evident that under certain specific icing conditions (such as those with an intermediate LWC value of 0.5 g/m^3^ indicated by the blue and green circle-dashed lines in Figure 13), the time required for the pitot tube to become fully obstructed by ice is longer than under other conditions. This results in a greater accumulation of ice within the pitot tube channel when it is blocked, leading to a more significant error at the moment of final obstruction compared to other conditions.(3)When the MVD remains constant, an increase in LWC results in a sharp rise in the rate of airspeed error. Conversely, maintaining a constant LWC, a rise in MVD causes only a slight increase in the rate of airspeed error. Notably, when LWC is high, this difference in speed becomes even less pronounced.

Meanwhile, as indicated by the data summary, should the pitot tube anti-icing system on the A320 aircraft malfunction during cruising:

Under normal icing conditions (when the MVD is within the range of 1–10 microns and the LWC is between 0.1–1.0 g/m^3^), when the aircraft pitot tube anti-icing system fails, the aircraft pitot tube may become blocked after 150 s of continuous icing, ceasing to function. Particularly, under icing conditions where LWC equals 1 g/m^3^, the aircraft pitot tube may become completely blocked and fail after 17 s of continuous freezing. In such a scenario, pilots and flight computers may lose the ability to accurately perceive the aircraft’s flight status, which could lead to incorrect operations and endanger flight safety. Therefore, establishing a “safe zone” to detect errors in airspeed data is crucial for ensuring the reliability and accuracy of flight information.

According to the national standard document HB8671-2020, titled “Specification for Reference Airspeed Tubes for Civil Aircraft Test Flights” (hereafter referred to as the “Specification”), the standard [26]:

The total pressure error (Cpt) of the aircraft pitot tube must adhere to the following specifications across the entire Mach number spectrum and throughout the full range of angles of attack within the flight envelope:(18)$$-\left(0.005+0.0005\left|A\right|\right)\leq C_{pt}\leq +-\left(0.005+0.0005\left|A\right|\right)$$
(19)$$C_{pt}=\frac{P_{ti}-P_{t}}{q_{c}}$$
(20)$$q_{c}=P_{t}-P_{s}$$
where:*C_pt_*—Non dimensional total pressure error;*P_ti_*—Total pressure sensed by aircraft pitot tube;*P_t_*—Total pressure of incoming flow;*P_s_*—Incoming static pressure;

By organizing the total pressure data errors from the aircraft pitot tube into an envelope surface, as depicted in Figure 14 (in which the green surface represents the allowed *C_pt_* upper limit, the blue surface represents the allowed *C_pt_* lower limits, and the red surface represents the actual *C_pt_*), and establishing the permissible error margins in accordance with the “specifications,” the safety envelope for environmental conditions during the aircraft’s cruising phase can be ascertained.

Notably, the area defined between the blue and green envelope surfaces during the aircraft’s cruising phase (where the red surface represents the total pressure error value of the aircraft pitot tube under varying icing conditions) encompasses the LWC icing time envelope surface for different MVD values.

Upon acquiring these necessary data, the aircraft will pre-load the icing conditions of the aircraft pitot tube into the flight computer. By detecting icing parameters in the environment using meteorological radar, temperature sensors, and other instruments, the flight warning computer will issue an ECAM warning when the safety threshold is approached. This alert informs the pilot that the aircraft pitot tube may freeze, potentially leading to airspeed sensing failure. Consequently, the pilot can proactively prepare to address the aircraft pitot tube malfunction and the disconnection of the autodrive system. This procedure aims to prevent accidents that could result from pilot or autodrive system errors due to incorrect airspeed data interpretation by the flight computer.

## 5. Conclusions

This article utilizes sophisticated numerical simulation methods to determine the correlation envelope surface that exists between icing conditions and airspeed data discrepancies identified by aircraft pitot tubes. The insights gleaned from this analysis lay a critical foundation for setting aircraft safety flight margins and defining the boundaries of environmental conditions, thereby ultimately guaranteeing the safety of commercial aviation operations. The comprehensive study yields the following enlightening conclusions:
(1)When LWC remains constant, as MVD increases, given the same icing duration, the airspeed error induced by the pitot tube icing exhibits an upward trend. However, the impact of water droplets and the icing process is inherently complex, involving a multi-parameter coupling in its numerical computation. Consequently, during the actual calculation process, it becomes evident that under certain specific icing conditions (such as those represented by the blue circle-solid line and blue circle-dashed line in Figure 13, where MVD is at an intermediate value of 35 microns), the time required for the pitot tube to become completely obstructed by ice is longer than under other conditions. This results in a greater accumulation of ice within the pitot tube channel when blocked, leading to a more significant error at the moment of complete blockage compared to other conditions. Simultaneously, with MVD held constant, the rate at which the airspeed error increases with rising LWC becomes notably steep.(2)When the MVD remains constant, an increase in LWC leads to a trend of escalating errors in airspeed measurements due to pitot tube icing over the same icing duration. Nevertheless, water droplet impingement and icing constitute a complex physical process, and its numerical simulation is a multi-parameter coupled process. Consequently, in practical calculations, it becomes evident that under certain specific icing conditions (such as those with an intermediate LWC value of 0.5 g/m^3^ indicated by the blue and green circle-dashed lines in Figure 13), the time required for the pitot tube to become fully obstructed by ice is longer than under other conditions. This results in a greater accumulation of ice within the pitot tube channel when it is blocked, leading to a more significant error at the moment of final obstruction compared to other conditions.(3)When the MVD remains constant, an increase in LWC results in a sharp rise in the rate of airspeed error. Conversely, maintaining a constant LWC, a rise in MVD causes only a slight increase in the rate of airspeed error. Notably, when LWC is high, this difference in speed becomes even less pronounced.(4)Should the anti-icing system of the A320 aircraft pitot tube fail while cruising in typical icing conditions, the aircraft pitot tube could become obstructed and cease to function after 150 s of uninterrupted exposure to icing. In the event of severe icing conditions, with an LWC of 1 g/m^3^, the aircraft pitot tube may suffer a complete blockage and failure after just 17 s of continuous freezing.(5)This article delineates the airspeed error LWC/icing time envelope surface for A320 aircraft during cruising conditions, considering ice particle sizes with MVD of 20, 35, and 50 microns.

## Figures and Tables

**Figure 1 sensors-24-07410-f001:**
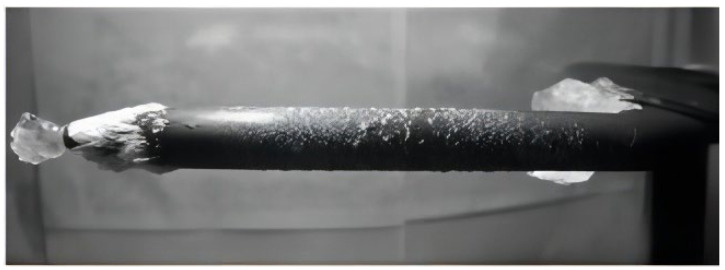
Ref. [2] Physical picture of aircraft pitot tube icing.

**Figure 2 sensors-24-07410-f002:**
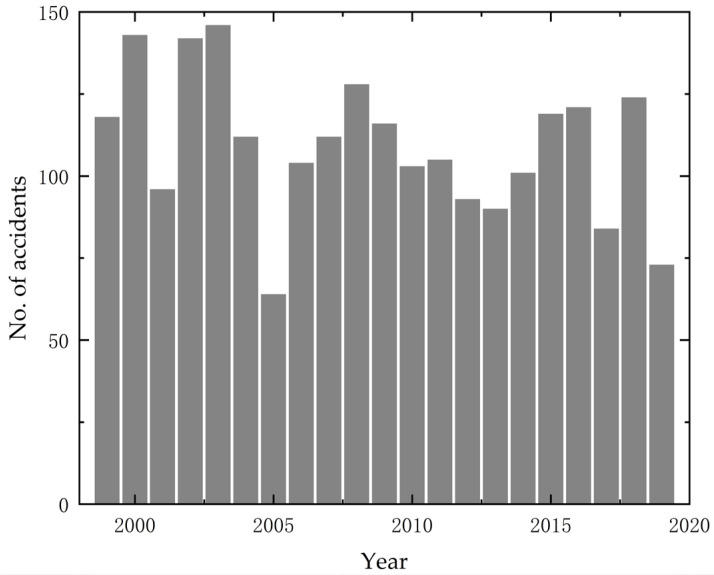
Statistics of Aircraft Icing Events from 2000 to 2020.

**Figure 3 sensors-24-07410-f003:**
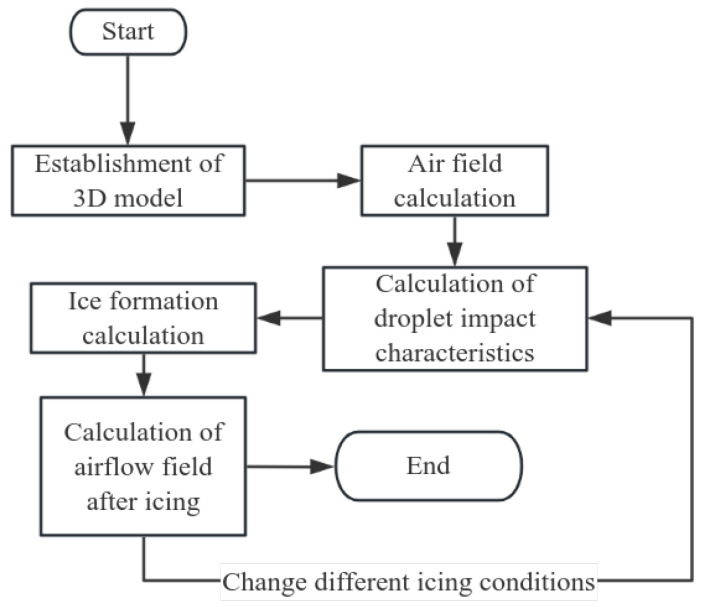
Numerical calculation process for icing.

**Figure 4 sensors-24-07410-f004:**
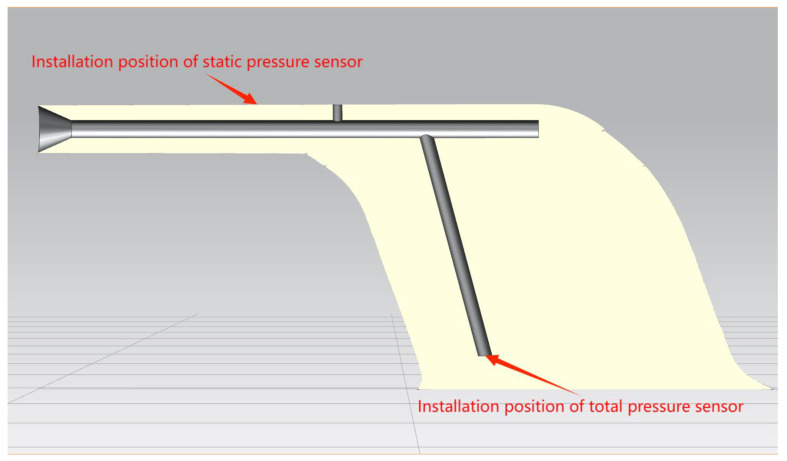
3D model of 0851HL.

**Figure 5 sensors-24-07410-f005:**
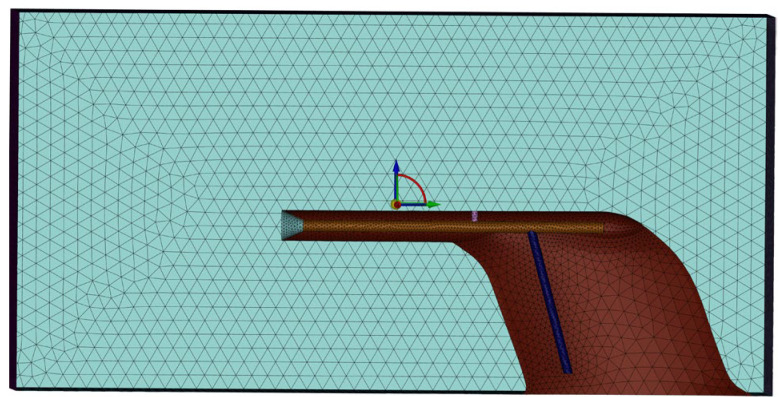
Fluid domain.

**Figure 6 sensors-24-07410-f006:**
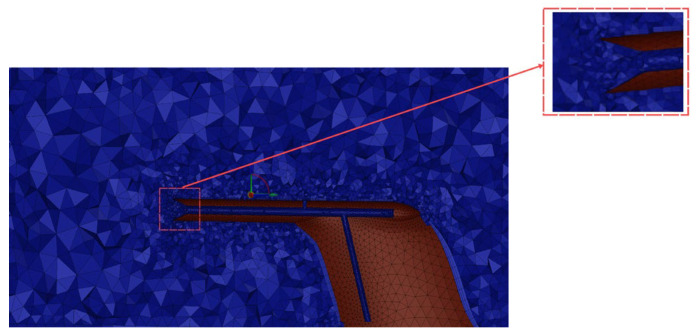
Fluid Computing Grids.

**Figure 7 sensors-24-07410-f007:**
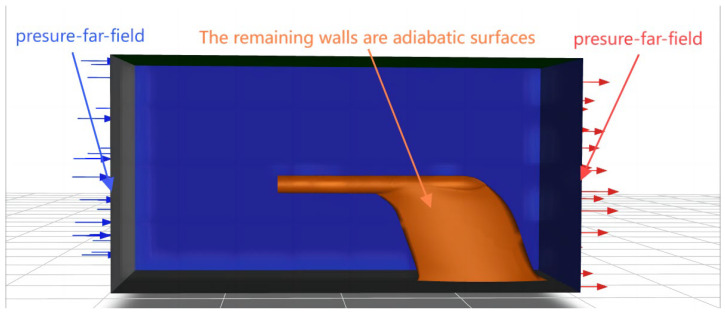
Boundary.

**Figure 8 sensors-24-07410-f008:**
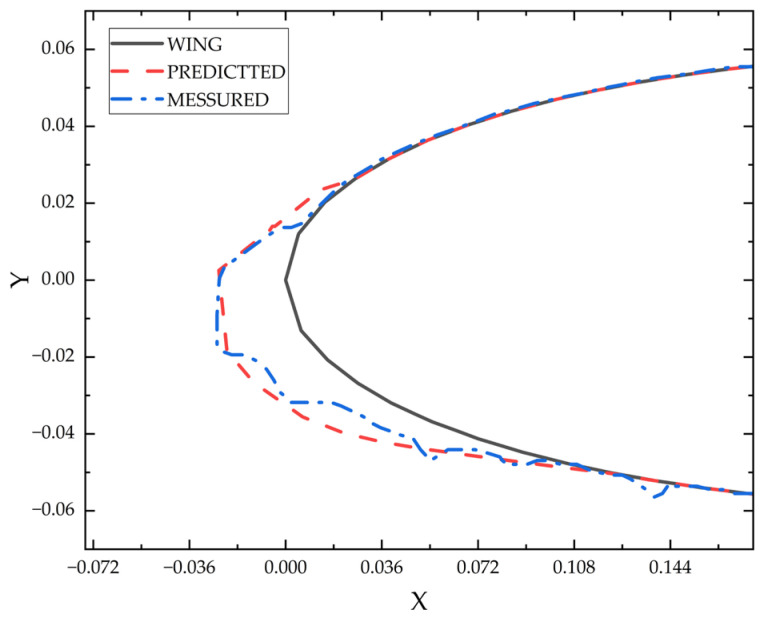
Ice formation at −28.3 °C.

**Figure 9 sensors-24-07410-f009:**
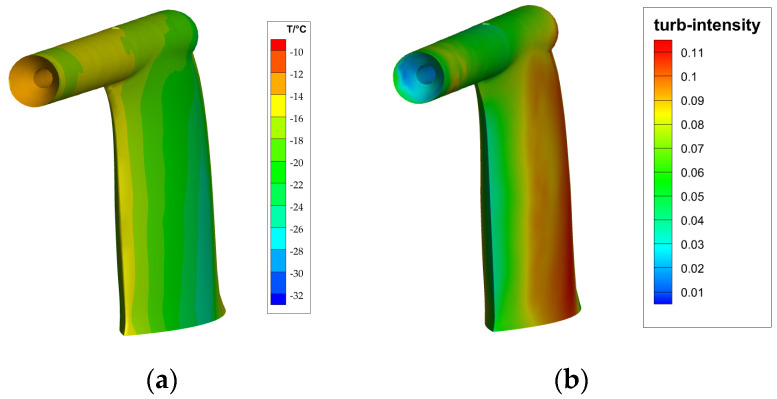
Results of airfield solution. (**a**) The contour of static temperature; (**b**) The contour of the turbulence intensity distribution.

**Figure 10 sensors-24-07410-f010:**
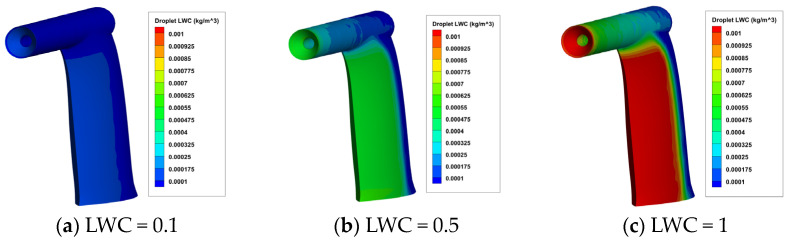
The contour of water droplet LWC on the surface of the aircraft pitot tube under different LWC with MVD = 20 microns.

**Figure 11 sensors-24-07410-f011:**
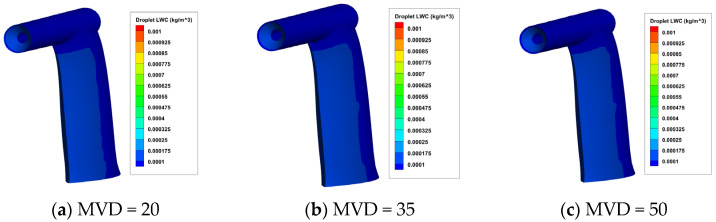
The contour of water droplet LWC on the surface of the aircraft pitot tube under different MVD with LWC = 0.1 g/m^3^.

**Figure 12 sensors-24-07410-f012:**
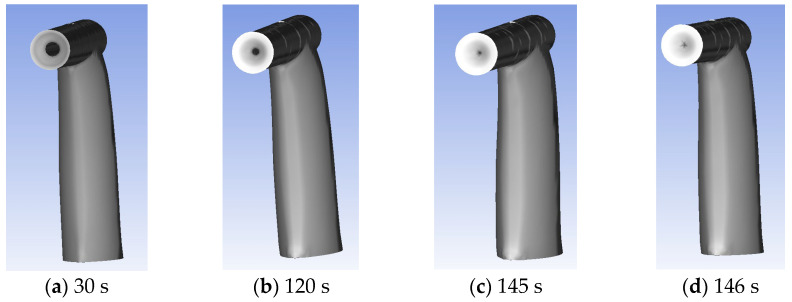
Aircraft pitot tube icing condition at MVD = 50 microns, LWC = 0.1 g/m^3^.

**Figure 13 sensors-24-07410-f013:**
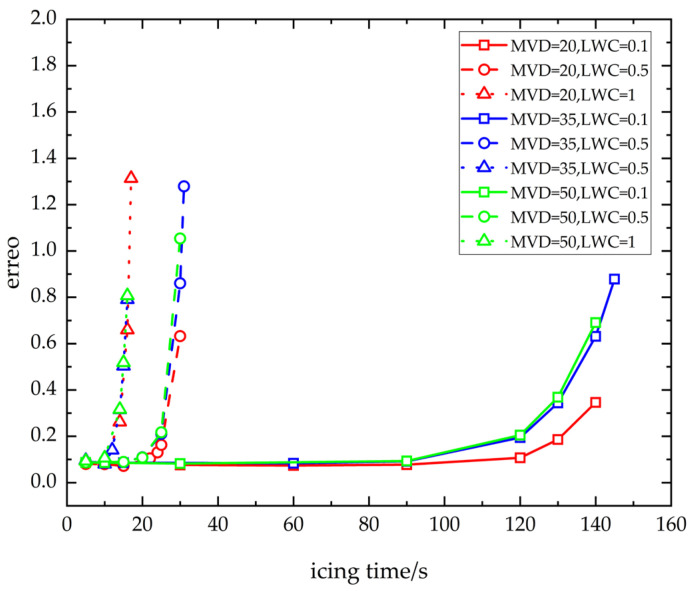
Error-icing time curve when MVD or LWC is constant.

**Figure 14 sensors-24-07410-f014:**
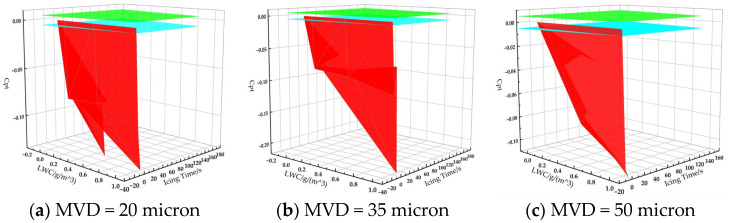
Safety zone of cruising state flight environment conditions.

**Table 1 sensors-24-07410-t001:** Boundary conditions for airfield calculation.

	Boundary Condition Type	Temperature/°C	Pressure/Pa	Incoming Mach Number
inlet	pressure-far-field	−41.15	33,161.2	0.752
outlet	pressure-far-field	−41.15	33,161.2	0.752

The thermal boundary condition of the aircraft pitot tube wall is adiabatic, with a surface roughness of 0.5 mm.

**Table 2 sensors-24-07410-t002:** Verify the calculation conditions of the example.

Pressure/Pa	Temperature/°C	Velocity/(m/s)	LWC/(g/m^3^)	MVD/Micron	a/°	Duration/s
101,300	−28.3	67.5	1	20	4	360

**Table 3 sensors-24-07410-t003:** A320Cruise state flight conditions.

Temperature T/°C	Pressure P/Pa	Mach Number Ma	Flight Altitude H/m
−40.25	33,161.2	0.752	8500

**Table 4 sensors-24-07410-t004:** Condition 1: MVD = 20 microns.

LWC/ (g/m^3^)	Icing Time/s							
0.1	30	60	90	120	130	140		
0.5	5	10	15	20	22	24	25	30
1	5	10	14	15	16	17		

(Note: When LWC is 0.1, 0.5, and 1 g/m^3^, and the freezing time is greater than 140, 30, and 17 s, respectively, the aircraft pitot tube will completely block, resulting in failure to obtain pressure data so that no further icing calculation will be carried out).

**Table 5 sensors-24-07410-t005:** Condition 2: MVD = 35 microns.

LWC/ (g/m^3^)	Icing Time/s							
0.1	5	15	60	90	120	130	140	145
0.5	5	10	15	20	25	30	31	
1	5	10	12	14	15	16		

(Note: When LWC is 0.1, 0.5, and 1 g/m^3^, and the freezing time is greater than 145, 31, and 16 s, respectively, the aircraft pitot tube will completely block, resulting in failure to obtain pressure data, so, no further icing calculation will be carried out).

**Table 6 sensors-24-07410-t006:** Condition 3: MVD = 50 microns.

LWC/ (g/m^3^)	Icing Time/s							
0.1	5	10	30	60	90	120	130	140
0.5	5	15	20	25	30			
1	5	10	14	15	16			

(Note: When LWC is 0.1, 0.5, and 1 g/m^3^, and the freezing time is greater than 140, 30, and 16 s, respectively, the aircraft pitot tube will completely block, resulting in failure to obtain pressure data, so the icing calculation will no longer be continued).

**Table 7 sensors-24-07410-t007:** Critical time for aircraft pitot tube icing and blockage.

MVD/Micron LWC/(g/m^3^)	20	35	50
0.1	160 s	146 s	145 s
0.5	31 s	32 s	31 s
1	20 s	17 s	17 s

## Data Availability

Data are contained within the article.

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
