# Peer review of "Numerical Simulation Study on Predicting the Critical Icing Conditions of Aircraft Pitot Tubes"

_sensors, 2024, doi:10.3390/s24227410_

Round 1
Reviewer 1 Report
Comments and Suggestions for Authors
Regarding the paper entitled "Numerical simulation study on predicting the critical icing conditions of aircraft pitot tubes", the concerns below should be satisfied in order to be possibly published:
- Figures, plots, and contours should be further illustrated, with better quality.
- There are many typos inside the manuscript. It seems that the authors haven't even revised it once before uploading the manuscript.
- The text must be improved, both in terms of demonstrations and mis-spellings (especially in the tables and figures, ...)
- The validations are seemingly non-realistic. Please provide proper CFD validation. How is that in all curves there are kinks while approaching to the experimental data!!
- The manuscript should be revised comprehensively to be published.
The text must be improved both in terms of demonstrations and mis-spellings (especially in the tables and figures, ...)
Author Response
For the reply, please check the attached file.

Reviewer 2 Report
Comments and Suggestions for Authors
Please check the attached file.

There are many grammatical and spelling errors in the manuscript. In some sentences, the words are confusingly used. It is recommended to significantly enhance the quality of English for the manuscript. Please refer to the attached file for comments in details.
Author Response
For the reply please check the attached file.

Round 2
Reviewer 2 Report
Comments and Suggestions for Authors_ About equations in section 2.1, the authors claim that "The equations display correctly when I read them, so I have uploaded the manuscript in PDF format. If there are still issues with the display of formulas, please check the PDF version of the manuscript." However, the equations in section 2.1 in PDF version are still overlapped with many strange symbols. The author should check the pdf version before uploading it for resubmission.
Comments on the Quality of English LanguageMinor editing of English language required.
Author Response
Comments 1: _ About equations in section 2.1, the authors claim that "The equations display correctly when I read them, so I have uploaded the manuscript in PDF format. If there are still issues with the display of formulas, please check the PDF version of the manuscript." However, the equations in section 2.1 in PDF version are still overlapped with many strange symbols. The author should check the pdf version before uploading it for resubmission.
Response 1: Thanks to the reviewer for suggestion. I revised the entire equations using MathType and uploaded the PDF in image format to address this issue. The image-based PDF file has been uploaded in the Non-published Material section. And I have made corrections to some details in the text and highlighted them.